# Immunogenicity of a Two-Dose Human Papillomavirus Vaccine Schedule in HIV-Infected Adolescents with Immune Reconstitution

**DOI:** 10.3390/vaccines10010118

**Published:** 2022-01-13

**Authors:** Supattra Rungmaitree, Charin Thepthai, Zheng Quan Toh, Noppasit Musiwiraphat, Alan Maleesatharn, Rattanachai Rermruay, Sathida Sungkate, Wanatpreeya Phongsamart, Keswadee Lapphra, Orasri Wittawatmongkol, Tararaj Dharakul, Kim Mulholland, Kulkanya Chokephaibulkit

**Affiliations:** 1Department of Pediatrics, Faculty of Medicine Siriraj Hospital, Mahidol University, Bangkok 10700, Thailand; sp.rt@live.com (S.R.); alan.mal@mahidol.ac.th (A.M.); wanatpreeya@gmail.com (W.P.); keswadee@gmail.com (K.L.); orasriw@hotmail.com (O.W.); 2Department of Immunology, Faculty of Medicine Siriraj Hospital, Mahidol University, Bangkok 10700, Thailand; charin.the@mahidol.ac.th (C.T.); tararaj.dha@mahidol.ac.th (T.D.); 3Infection and Immunity, Murdoch Children’s Research Institute, Parkville, VIC 3052, Australia; zheng.quantoh@mcri.edu.au (Z.Q.T.); kim.mulholland@lshtm.ac.uk (K.M.); 4Department of Pediatrics, The University of Melbourne, Parkville, VIC 3010, Australia; 5Siriraj Institute of Clinical Research, Faculty of Medicine Siriraj Hospital, Mahidol University, Bangkok 10700, Thailand; noppasit.mus@mahidol.ac.th; 6Department of Pediatrics, Bhumibol Adulyadej Hospital, Bangkok 10220, Thailand; rattanachai_r@taf.mi.th; 7Department of Pediatric, Maharat Nakhon Ratchasima Hospital, Nakhon Ratchasima 30000, Thailand; Sathy_skx@hotmail.com; 8Department of Infectious Disease and Epidemiology, London School of Hygiene and Tropical Medicine, London WC1E 7HT, UK

**Keywords:** HPV vaccine, HIV adolescents, Thai adolescents, HPV in HIV, two-dose schedule

## Abstract

HIV-infected patients are at increased risk of human papillomavirus (HPV) acquisition and HPV-associated diseases. This study set out to determine whether a two-dose (2D) HPV vaccination schedule was sufficient in HIV-infected adolescents with immune reconstitution (IR) following antiretroviral treatment. Participants aged 9–15 years who had CD4 cell counts > 500 cells/mm^3^ and HIV-1 RNA < 40 copies/mL for at least one year were assigned to the 2D schedule, while older participants or those without IR received a three-dose (3D) schedule. Antibodies to HPV-16 and -18 were measured using a pseudovirion-based neutralization assay. A total of 96 subjects were enrolled; 31.3% and 68.7% received the 2D and 3D schedule, respectively. Of these, 66.7% and 57.6% of the 2D and 3D participants, respectively, were male. The seroconversion rates for HPV-16 and HPV-18 were 100% in all cases, except for HPV-18 in males who received the 3D schedule (97.4%). In males, the anti-HPV-16 geometric mean titers (GMTs) were 6859.3 (95% confidence interval, 4394.3–10,707.1) and 7011.1 (4648.8–10,573.9) in the 2D and 3D groups (*p* = 0.946), respectively, and the anti-HPV-18 GMTs were 2039.3 (1432.2–2903.8) and 2859.8 (1810.0–4518.4) in the 2D and 3D (*p* = 0.313) groups, respectively. In females, the anti-HPV-16 GMTs were 15,758.7 (8868.0–28,003.4) and 26,241.6 (16,972.7–40,572.3) in the 2D and 3D groups (*p* = 0.197), respectively, and the anti-HPV-18 GMTs were 5971.4 (3026.8–11,780.6) and 9993.1 (5950.8–16,781.1) in the 2D and 3D groups (*p* = 0.271), respectively. In summary, a 2D schedule is as immunogenic in young adolescents with IR as a 3D schedule in older subjects and those without IR.

## 1. Introduction

Similar to adults, adolescents living with human immunodeficiency virus (ALHIV) have a higher incidence of human papillomavirus (HPV) infection, abnormal pap smears, and persistent HPV infection, leading to an increased risk of anogenital cancers than non-ALHIV [1,2,3]. The current evidence has demonstrated that the HPV vaccine is safe and efficacious for people living with HIV, and they should therefore be vaccinated with the HPV vaccine [4].

Three prophylactic HPV vaccines, bivalent (bHPV, Cervarix^®^), quadrivalent (qHPV, Gardasil^®^) and nonavalent (9-valent, Gardasil^®^ 9), have been licensed. All three vaccines exhibited excellent safety profiles and were highly efficacious in preventing HPV infections and premalignant anogenital lesions caused by the HPV types included in the vaccines [5,6,7,8]. The Advisory Committee on Immunization Practices Recommendations for Vaccination (ACIP) recommended HPV vaccination for all persons aged up to 26 years, and the recommended dosing schedule depends on the age of the patient at vaccine initiation; a two-dose (2D) schedule should be given to adolescents aged 9–15 years, and a three-dose (3D) schedule, to individuals aged 15 years or older [9,10]. The 2D HPV vaccine schedule was introduced based on non-inferior antibody responses observed between healthy young adolescents aged 9–14 years and young adults aged 15–25 years [11,12]. However, for immunocompromised individuals, including those who are infected with HIV, the 3D schedule is recommended regardless of their age [9,10].

HPV vaccines have been shown to induce humoral and cellular immune responses against vaccine genotypes. T-memory cells play a significant role in long-term protection, particularly when the antibody has waned. The HPV vaccine is thought to be less immunogenic and efficacious in HIV-infected individuals due to the immunodeficient state compared to healthy individuals. However, effective antiretroviral therapy (ART) has resulted in normalized immune function and CD4 T-lymphocyte (CD4) counts. A study in ALHIV with CD4 counts more than 350 cells/mm^3^ found similar immunogenicity with a qHPV vaccine compared to that in healthy adolescents [13]. The question of whether a 2D HPV vaccine schedule is immunogenic in HIV-infected individuals with successful ART under the age of 15 years old remains to be answered.

We conducted a cohort study to investigate whether young adolescents with successful ART who received a 2D HPV vaccine schedule generated a similar HPV antibody response to older adolescents or those subjects without immune reconstitution who received a 3D schedule.

## 2. Methods and Participants

### 2.1. Study Participants and Design

This single-center prospective cohort study enrolled HIV-infected adolescents aged 9–24 years at the Pediatric HIV Clinic at Siriraj Hospital, a tertiary care center in Bangkok, Thailand. After the informed consent process, the participants were assigned to receive either a 2D or 3D schedule of the HPV vaccine, according to age and HIV immune status. Participants aged 9–15 years who had been receiving ART with immune reconstitution, defined as CD4 counts >500 cells/mm^3^ and HIV-1 RNA levels (viral loads; VLs) < 40 copies/mL for at least one year, were assigned to the 2D schedule. Older adolescents (15–24 years old) or young participants who did not meet the criterion of immune reconstitution were assigned to the 3D schedule. The 2D group received HPV vaccinations scheduled at months 0 and 6, and the 3D group received vaccinations scheduled at months 0, 1–2, and 6. All the male participants received the qHPV vaccine, and the females received bHPV, according to the local guidelines and vaccine availability at the time of the study’s initiation. Participants who had previously received the HPV vaccine or received immunosuppressive agents were excluded from the study.

Blood samples (5 mL) were collected at pre-vaccination (month 0) and at 1 to 3 months after the last dose (months 7–9). Any adverse events following immunization were observed according to routine practice. This study was approved by the local Institutional Review Board.

### 2.2. Immunogenicity Assessments

Antibodies to HPV-16 and -18 were measured using the HPV pseudovirion-based neutralization assay (PBNA), as described by Barzon et al. [14]. The neutralizing titer (ED_50_) is defined as the highest serum dilution that reduces the secreted alkaline phosphatase activity by at least 50% in comparison to a control (pseudovirion without serum). A sample with an ED_50_ value ≥ 100 was considered HPV-seropositive. Seronegative titers (<100) were given a value of 50. All the laboratory staff were blinded to the vaccination status of each participant, and each sample was identified according to a unique study number.

### 2.3. Statistical Analysis

Demographic data including age, CD4 counts, and VL are summarized by descriptive statistics. Continuous data are presented using medians, interquartile ranges (IQRs), and proportions, as appropriate. Differences between participants receiving 2D or 3D schedules in terms of demographic characteristics were compared using the Mann–Whitney U test for continuous variables and a χ^2^ test or Fisher’s exact test for category variables. Seroconversion was defined as a change from seronegative results at pre-vaccination (month 0) to seropositive results at 1 to 3 months after the last dose (months 7–9). HPV type-specific geometric mean titers (GMTs) were calculated. We log-transformed the neutralizing antibody titers and compared the 2D and 3D schedule groups by using Student’s t-test. All the statistical analyses were performed using STATA 15.1 (StataCorp, College Station, TX, USA). A *p* value < 0.05 was considered statistically significant for all the analyses.

## 3. Results

A total of 96 adolescents were enrolled, of which 93 (96.9%) were perinatally infected, and 58 (60.4%) were male. Of the 30 (31.3%) and 66 (68.7%) participants who were assigned to the 2D and 3D groups, 20 (66.7%) and 38 (57.6%) were males, respectively.

According to the demographic data (Table 1), there were no significant differences between the 2D and 3D groups for both sexes with respect to the World Health Organization HIV clinical disease stage. The participants (both sexes) who received the 2D schedule had significantly higher levels of current and nadir CD4 cells compared to those who received the 3D schedule. The proportion of participants who received the non-nucleoside reverse transcriptase inhibitor (NNRTI) regimen was found to be higher among those who received the 2D schedule compared to those who received the 3D schedule, although this was only significant in the female cohort (Table 1).

All 58 male participants who received the qHPV vaccine were seronegative for HPV-16 and -18 at pre-vaccination. The seroconversion rates for HPV-16 were 100% (20/20) and 100% (38/38) in the 2D and 3D groups, respectively, and the seroconversion rates for HPV-18 were 100% (20/20) and 97.4% (37/38) in the 2D and 3D groups, respectively. Of the 38 female participants who received bHPV, two were seropositive for HPV-16 and -18 at pre-vaccination (one for HPV-16 and one for HPV-18). All the female participants who were initially seronegative for an HPV serotype at baseline demonstrated 100% seroconversion for both HPV-16 and -18 in both the 2D and 3D groups (Table 2 and Figure 1).

In the male participants, the neutralizing anti-HPV-16 antibody geometric mean titers (GMTs) were 6859.3 (95% CI: 4394.3–10,707.1) and 7011.1 (95% CI: 4648.8–10,573.9) in the 2D and 3D groups (*p* = 0.946), respectively, and the anti-HPV-18 antibody GMTs were 2039.3 (95% CI: 1432.2–2903.8) and 2859.8 (95%CI: 1810.0–4518.4) in the 2D and3D groups (*p* = 0.313), respectively. In the female participants, the anti-HPV-16 antibody GMTs were 15,758.7 (95% CI: 8868.0–28,003.4) and 26,241.6 (95% CI: 16,972.7–40,572.3) in the 2D and 3D groups (*p* = 0.197), respectively, and the anti-HPV-18 antibody GMTs were 5971.4 (95% CI: 3026.8–11,780.6) and 9993.1 (95% CI: 5950.8–16781.1) in the 2D and 3D groups (*p* = 0.271), respectively. There was no difference in GMT between the 2D and 3D groups for each of the vaccines. The GMT in males who received qHPV was about 40% lower for both HPV-16 and -18 compared to that in the females who received bHPV (Table 2). There was also no correlation between the CD4 cell count and anti-HPV-16 and -18 antibody GMT response after vaccination (Figure 2).

## 4. Discussion

Human papillomavirus vaccination is strongly recommended for ALHIV individuals because they have a higher risk of acquiring HPV infections and lower HPV clearance rates compared to HIV-uninfected individuals, thereby leading to an increased risk of HPV-associated diseases such as anogenital cancers [15,16,17,18]. However, the cost and logistical constraints of delivering HPV vaccines in resource-limited settings have hindered access to the vaccines, making the highest-risk population, that with ALHIV, vulnerable to HPV-associated diseases. This study examined whether a 2D HPV vaccine schedule in HIV-infected adolescents aged 9–15 years who received ART with immune reconstitution would generate a similar HPV antibody response to older adolescents or those without immune reconstitution who received a 3D schedule. To our knowledge, this is the first study to demonstrate similar antibody responses between HIV-infected adolescents who received a 2D HPV vaccine schedule and HIV-infected older adolescents who received a 3D schedule. Our findings support a formal evaluation of the 2D schedule for HIV-infected individuals receiving successful ART with immune reconstitution. The 2D schedule will reduce costs and alleviate the logistical constraints associated with vaccine delivery, leading to improved access to HPV vaccination for adolescents aged 9–15 years, especially in resource-poor settings with high burdens of HIV infections.

The seroconversion rate for HIV-infected adolescents in our cohort was 97–100% for both HPV-16 and -18 in both the 2D and 3D groups, consistent with previous studies reported for ALHIV individuals who received a 3D schedule [19,20,21]. The neutralizing antibody GMTs for HPV-16 and -18 in our female participants receiving a 3D schedule with the bHPV vaccine were consistent with the levels reported for HIV-infected 15–25-year-old adolescents and women in a previous cohort study [21], providing confidence in our data. In general, we found non-statistically lower titers in the 2D group compared to the 3D group, and the difference was more prominent in females receiving bHPV. Despite this, it is important to note that the clinical relevance of this lower antibody response remains unknown. Furthermore, the small sample size and the timing of blood collections following the last vaccine dose (1–3 months, due to feasibility of sample collection) may have contributed towards the variations of the antibody response in both groups. A larger study comparing 2D and 3D schedules in HIV-infected adolescents and young adults will be needed to confirm our findings.

HIV infects and deletes CD4+ T cells that normally support adaptive T- and B-cell responses including high-quality antibody production [22]. In previous studies, the antibody titer for type-specific HPV was lower in HIV-infected patients compared to uninfected patients, particularly in those with lower CD4 levels [13,19,20,21,23]. We sought to determine whether there was a relationship between the number of CD4+ T cells and antibody GMT response following HPV vaccination. Interestingly, no correlation was observed. This may be due to patients with lower CD4 levels in our cohort receiving a 3D HPV vaccination schedule, which might have compensated for the antibody response associated with a lower CD4+ cell count. Conversely, this suggests that a CD4 level > 500 cells/mm^3^ is an appropriate marker of immune reconstitution, which may be immunogenic following a 2D HPV vaccination schedule.

Another finding of this study in both the 2D and 3D groups, in line with other studies, was the superior overall immunogenic response for bHPV compared to qHPV for both HPV-16 and-18 [21,24]. We found that the neutralizing antibody levels were 2.3-fold higher with bHPV compared to qHPV for HPV-16, and 2.9-fold higher for HPV-18 in the 2D group. Similarly, in the 3D group, the anti HPV-16 neutralizing antibody levels were 3.7-fold higher with bHPV compared to qHPV, and 3.5-fold higher for HPV-18. As the vaccine used in our study was gender-specific according to local guidelines, bHPV for females and qHPV for males, females were found to have higher antibody levels than males. The adjuvant system 04 (AS04) in bHPV is thought to play a key role in determining the differences in immunogenicity and efficacy profiles between the two vaccines [25,26]. The more immunogenic vaccine has the potential to induce long-lasting protection and would be preferable, particularly among these high-risk populations. However, studies of the magnitude and persistence of vaccine-induced HPV antibodies in HIV-infected patients have, thus far, been limited, with the longest follow-up documented being 4 years post-vaccination [27]. Additionally, the clinical significance of higher antibody concentrations is unknown, as there is no established threshold correlating with efficacy in any population studied to date.

There are some limitations in this study. The relatively small number of participants in each group might have prevented us from detecting any significant differences in antibody levels. In addition, we did not include a young healthy control group, which meant that a comparison of HPV vaccine immunogenicity between ALHIV and healthy volunteers cannot be made. We are also unable to compare immunogenicity between bHPV and qHPV in the same sex, as we used a gender-specific vaccine type. Finally, the follow-up in our study was limited, and we did not examine the clinical efficacy of the vaccine. Further studies are needed with larger numbers of participants in more settings and a longer follow-up period to determine the duration of the immunity as well as the clinical efficacy in ALHIV.

## 5. Conclusions

Adolescents aged 9–15 years with immune reconstitution, indicated by CD4 counts > 500 cells/mm^3^ and persistent virologic suppression, can generate a similar antibody response following a 2D HPV vaccination schedule when compared with older adolescents and those without immune reconstitution who receive a 3D schedule. Further studies to formally evaluate the 2D HPV vaccination schedule in HIV-infected adolescents are warranted. Our findings are particularly relevant for countries where HIV infections and HPV diseases are prevalent and where HPV vaccine coverage is low.

## Figures and Tables

**Figure 1 vaccines-10-00118-f001:**
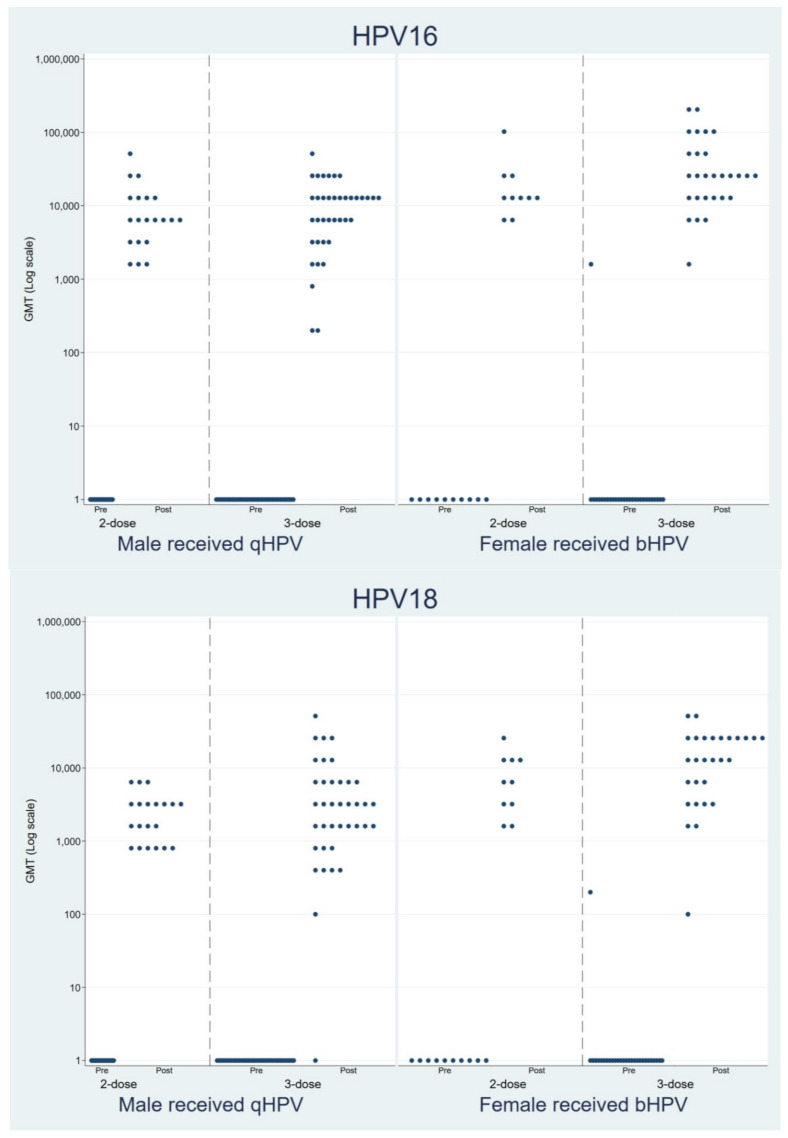
The geometric mean titers of human papillomavirus 16 and 18 among the 2- and 3-dose groups, Cervarix^®^ and Gardasil^®^ at pre-vaccination, and at 1 to 3 months after the last dose.

**Figure 2 vaccines-10-00118-f002:**
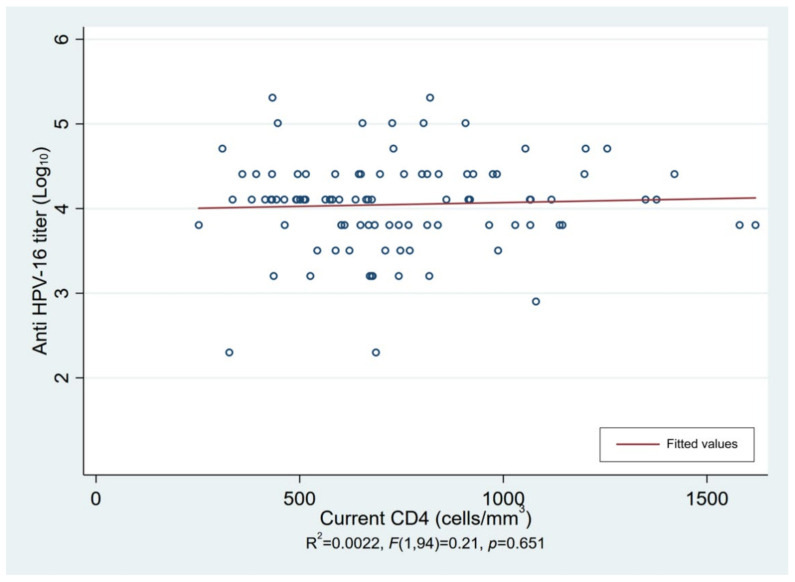
The correlation between CD4 cell count and anti-HPV-16 and -18 antibody response after vaccination.

**Table 1 vaccines-10-00118-t001:** Demographic characteristics of participants by vaccination schedule.

	Male				Female			
Characteristics	Total (*n* = 58)	2-Dose ^a^(*n* = 20)	3-Dose ^a^(*n* = 38)	*p*-Value	Total (*n* = 38)	2-Dose ^a^(*n* = 10)	3-Dose ^a^(*n* = 28)	*p*-Value
Age; *n* (%)								
≤15 years	27 (46.5)	20 (100.0)	7 (18.4)	<0.001	11 (28.9)	10 (100.0)	1 (3.6)	<0.001
>15 years	31 (53.5)	-	31 (81.6)		27 (71.1)	-	27 (96.4)	
Age at ART start; Median (IQR)	3.4 (1.0–7.7)	1.0 (0.3–3.0)	5.6 (1.4–9.7)	<0.001	3.2 (0.6–8.2)	1.6 (0.3–5.5)	4.0 (0.7–9.4)	0.131
Clinical stage								
Worst WHO stage; *n* (%)								
Stage 1	5 (8.6)	3 (15.0)	2 (5.3)	0.251	4 (10.5)	2 (20.0)	2 (7.1)	0.785
Stage 2	14 (24.1)	6 (30.0)	8 (21.0)		6 (15.8)	1 (10.0)	5 (17.9)	
Stage 3	22 (37.9)	8 (40.0)	14 (36.8)		15 (39.5)	4 (40.0)	11 (39.3)	
Stage 4	17 (29.3)	3 (15.0)	14 (36.8)		13 (34.2)	3 (30.0)	10 (35.7)	
CD4								
Nadir CD4 cells/mm^3^; Median (IQR)	362 (21–572)	615 (354–774)	241 (9–414)	<0.001	198 (33–426)	531 (287–963)	87 (30–337)	0.003
At enrollment; Median (IQR)	626 (501–899)	866 (639–1061)	541 (432–743)	<0.001	700 (525–1008)	1152 (702–1359)	599 (487–746)	0.009
≤500; *n* (%)	14 (24.1)	-	14 (36.8)	0.001	7 (18.4)	-	7 (25.0)	0.156
>500; *n* (%)	44 (75.9)	20 (100.0)	24 (63.2)		31 (81.6)	10 (100.0)	21 (75.0)	
VL; *n* (%)								
<50 copies/mL	49 (84.5)	20 (100.0)	29 (76.3)	0.021	34 (89.5)	10 (100.0)	24 (85.7)	0.556
≥50 copies/mL	9 (15.5)	-	9 (23.7)		4 (10.5)	-	4 (14.3)	
Duration of VL < 50 copies/mL (months); Median (IQR)	91.8 (81.9–96.6)	90.0 (77.5–94.3)	92.0 (85.3–102.3)	0.330	56.7 (9.4–86.2)	72.5 (32.6–87.4)	54.3 (6.6–82.7)	0.136
Baseline ART Regimen; *n* (%)								
NNRTI	33 (56.9)	14 (70.0)	19 (50.0)	0.172	23 (60.5)	10 (100.0)	13 (46.4)	0.003
PI	25 (43.1)	6 (30.0)	19 (50.0)		15 (39.5)	-	15 (53.6)	

^a^ The 2-dose group received HPV vaccinations scheduled at months 0 and 6, while the 3-dose group received vaccinations scheduled at months 0, 1–2, and 6. Abbreviations: ART, antiretroviral treatment; IQR, interquartile range; CD4, CD4 T lymphocyte; VL, viral load (HIV RNA level); NNRTI, non-nucleoside reverse transcriptase inhibitor-based regimens; PI, protease inhibitor-based regimens.

**Table 2 vaccines-10-00118-t002:** Seroconversion and geometric mean titers (GMTs) with 95% confidence intervals (95%CIs) for HPV-16 and -18 in males and females by vaccination schedule.

	2 Doses ^a^	3 Doses ^a^	*p*-Value
Male (Gardasil^®^)			
Type 16			
Seroconversion; *n* (%)	20/20 (100.0)	38/38 (100.0)	-
GMT (95%CI)	6859.3 (4394.3–10,707.1)	7011.1 (4648.8–10,573.9)	0.946 ^b^
Type 18			
Seroconversion; n (%)	20/20 (100.0)	37/38 (97.4)	1.000
GMT (95%CI)	2039.3 (1432.2–2903.8)	2859.8 (1810.0–4518.4)	0.313 ^b^
Female (Cervarix^®^)			
Type 16			
Seroconversion; n (%)	10/10 (100.0)	27/27 (100.0)	-
GMT (95%CI)	15,758.7 (8868.0–28,003.4)	26,241.6 (16,972.7–40,572.3)	0.197 ^b^
Type 18			
Seroconversion; *n* (%)	10/10 (100.0)	27/27 (100.0)	-
GMT (95%CI)	5971.4 (3026.8–11,780.6)	9993.1 (5950.8–16,781.1)	0.271^b^

^a^ The 2-dose group received a HPV vaccination scheduled at months 0 and 6, and the 3-dose group received a vaccination that was scheduled at months 0, 1–2 and 6; ^b^ The *p*-value of the geometric mean titers (GMTs) was calculated using the t-test with log-transformed data of the GMTs.

## Data Availability

Data sharing is not applicable to this article.

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
