# Peer review of "Immunogenicity of a Two-Dose Human Papillomavirus Vaccine Schedule in HIV-Infected Adolescents with Immune Reconstitution"

_vaccines, 2022, doi:10.3390/vaccines10010118_

Round 1

Reviewer 1 Report

The manuscript by Rungmaitree et al. compared the immunogenicity caused by HPV vaccines in young adolescents with 2D and those older or without IR with 3D administrations. They found there is no statistic difference between these groups, demonstrating 2D is enough for young HIV-infected patients with successful ART. Their study is extremely helpful for regions with high HIV incidences and limited HPV vaccine supply. Also they provided enough details for the experiments and expressed the main ideas very clearly. However, I would like to suggest several minor revisions to meet the requirement of Vaccines. 

1. To make their points clearer, they should mention relationship between HIV and HPV infections at the first beginning of the abstract. 

2. Since the authors mentioned the male ratio in total in abstract, they should also give the information for the ratio in 2D and 3D, respectively.  

3. Please use full name of the profession terms at their first appearances, such as “GMT”. 

4. The anti-HPV GMT differences are really obvious between male and female. Though they argued it may be cause by different vaccines for the groups, I’m curious whether other similar studies observed the same trend. 

5. As they also partially claimed, more studies should be warranted to make the conclusion more convincing and solid in the future: the priority is the measurements in the long time points; expand the participants to a larger number and also different regions; more controls should be set up properly; more immune response parameters should also be considered; most importantly, the final clinical output should be carefully followed and recorded at a long time period. 

Author Response

Responses to the comments of Reviewer # 1

Comment: 1) To make their points clearer, they should mention relationship between HIV and HPV infections at the first beginning of the abstract.

Response: Thank you for pointing this out. We have added the relationship between HIV and HPV infections at the first beginning of the abstract. (Highlighted, line 21)

Comment: 2) Since the authors mentioned the male ratio in total in abstract, they should also give the information for the ratio in 2D and 3D, respectively. 

Response: We added the information for the male ratio in 2D and 3D in the abstract. (Highlighted, line 27-28)

Comment: 3) Please use full name of the profession terms at their first appearances, such as “GMT”.

Response: We have added a full name of GMT in the abstract. (Highlighted, line 29)

Comment: 4) The anti-HPV GMT differences are really obvious between male and female. Though they argued it may be cause by different vaccines for the groups, I’m curious whether other similar studies observed the same trend.

Response: Other previous studies found the same trend that the bHPV (AS04-HPV-16/18) induced higher immunogenicity compared to qHPV (HPV-6/11/16/18), observed in both healthy women and women living with HIV (reference 21, 24 in the manuscript). However, we are unable to compare immunogenicity between bHPV and qHPV in the same sex as we used gender-specific vaccine type in accordance with the National Guidelines and regulatory authority approval.

Comment: 5) As they also partially claimed, more studies should be warranted to make the conclusion more convincing and solid in the future: the priority is the measurements in the long time points; expand the participants to a larger number and also different regions; more controls should be set up properly; more immune response parameters should also be considered; most importantly, the final clinical output should be carefully followed and recorded at a long time period.

Response: We incorporated your suggestion in the ‘Discussion’ section. (Highlighted, line 224-226)

Reviewer 2 Report

The manuscript determined whether 2-dose HPV vaccination is sufficient in HIV-infected adolescents aged 9-15 years with immune reconstitution. Their results showed that a 2-dose HPV vaccination is immunogenic in young adolescents with immune reconstitution after comparing it to a 3-dose HPV vaccination in older adolescents or those without immune reconstitution. These findings may benefit the relevant countries where HPV and HIV-associated diseases are prevalent if they could validate the results in more participants.

However, I’m curious about why the participants with 2-dose HPV vaccination have more CD4 cells when compared to those who received 3-dose vaccination in Table 1 (866 to 541 in male, 1152 to 599 in female). How do the authors explain this?

Furthermore, the authors summarized 2-dose HPV vaccination may be sufficient in HIV-infected adolescents aged 9-15 years with immune reconstitution because of the seropositive response, so is there any significant difference between 2-dose and 3-dose HPV vaccination in the specific adolescents? What’s the disadvantage of the 2-dose HPV vaccination when compared to the 3-dose HPV vaccination based on their results?

Author Response

Responses to the comments of Reviewer # 2

Comment: 1) However, I’m curious about why the participants with 2-dose HPV vaccination have more CD4 cells when compared to those who received 3-dose vaccination in Table 1 (866 to 541 in male, 1152 to 599 in female). How do the authors explain this?

Response: We allocated participants to receive either 2D schedule or 3D schedule of HPV vaccine according to age and HIV immune status. The participants 9-15 years of age who had been receiving ART with immune reconstitution (IR); defined as CD4 counts > 500 cells/mm3 and HIV-1 RNA level (viral load; VL) < 40 copies/mL for at least one year, were assigned to the 2D schedule. The older adolescents (15-24 years old) or participants who did not meet the criteria of IR were assigned to the 3D schedule. The participants who had lower CD4 count were not likely to meet IR criteria would be receiving 3D schedule.

Comment: 2) Furthermore, the authors summarized 2-dose HPV vaccination may be sufficient in HIV-infected adolescents aged 9-15 years with immune reconstitution because of the seropositive response, so is there any significant difference between 2-dose and 3-dose HPV vaccination in the specific adolescents? What’s the disadvantage of the 2-dose HPV vaccination when compared to the 3-dose HPV vaccination based on their results?

Response: We are unable to identify the difference of 2D and 3D in any specific group. The only disadvantage of the 2D schedule compared to 3D was the insignificantly lower neutralizing GMT which may not have clinical significance.

Reviewer 3 Report

The manuscript submitted by Rungmaitree et al, entitled "Immunogenicity of a 2-Dose Human Papillomavirus Vaccine Schedule in HIV-infected Adolescents with Immune Reconstitution", aims to verify if a 2D HPV vaccine schedule in HIV-infected adolescents aged 9-15 years, who received ART with immune reconstitution, will generate a similar HPV antibody response as older adolescents or those without immune reconstitution who received 3D. The data present in this study is very novel and relevant for this field, since is the first study to demonstrate similar antibody responses between HIV-infected adolescents who received 2D of HPV vaccine and HIV-infected older adolescents who received 3D. The results, figures and tables are clear and the manuscript is written in a very comprehensive manner, allowing a very easy reading. Additionally, the results are very well discussed, supporting a solid and relevant conclusion. However, in the opinion of this reviewer, authors should improve/discuss some points before its acceptance for publication:

1- The role of T-memory cells in vaccination should be briefly presented in the Introduction section.

2- The differences in the mean titer of human papillomavirus 16 and 18 among 2-dose and 3-dose between the genders should be more discuss.

3- Please comment why the p-values in the figure 2 are so higher.

4- If possible, it will be great if authors can add some data about the dynamics of serum IgM and IgG levels.

5- Did author have any information about the vaccine efficiency in clinical terms?

Author Response

Responses to the comments of Reviewer # 3

Comment: 1) The role of T-memory cells in vaccination should be briefly presented in the Introduction section.

            Response: We have added the role of T-memory cells in vaccination to emphasize this point in the ‘introduction’ section. (Highlighted, line 59-61)

Comment: 2) The differences in the mean titer of human papillomavirus 16 and 18 among 2-dose and 3-dose between the genders should be more discuss.

Response: The difference between genders was added in the discussed (line 207-209). That paragraph discussed the difference of immunogenicity between the two vaccines and is the reason for the difference of GMT between genders due to the use of gender-specific study vaccine.

Comment: 3) Please comment why the p-values in the figure 2 are so higher.

Response: The figure 2 revealed the absence of correlation between CD4 count and the GMT (high p-value). We commented this in the discussion (line 196-199). The patients with lower CD4 levels in our cohort receiving 3D of HPV vaccine, that might have compensated the antibody response associated with lower CD4+ cell count.

Comment: 4) If possible, it will be great if authors can add some data about the dynamics of serum IgM and IgG levels.

Response: We agree with the comment. However, we did not have a follow-up study to measure the antibody, and therefore the data was unavailable.

Comment: 5) Did author have any information about the vaccine efficiency in clinical terms?

 Response: You have raised an important point here. Unfortunately, we do not have an efficacy data. Long-term follow-up study is needed as raised in the limitations.
